# Spatiotemporal Analysis of Urban Expansion in the Mountainous Hindu Kush Himalayas Region

Zhenhua Chao [1], Zhanhuan Shang [2], Chengdong Fei [1,*], Ziyi Zhuang [1] and Mengting Zhou [1]

1   School of Geographic Science, Nantong University, Nantong 226007, China
2   College of Ecology, Lanzhou University, Lanzhou 730000, China
*   Correspondence: 2021110004@stmail.ntu.edu.cn

**Abstract:** As a major human activity, urbanization exerts a strong impact on the fragile ecosystem in the Hindu Kush Himalayas (HKH) region. To maintain sustainable development, reliable data on urban land change are required to assess the impact of urbanization. Here, the reliability evaluation of four global fine-resolution impervious surface area (ISA) products: global annual impervious area (GAIA), global annual urban dynamics (GAUD), global impervious surface area (GISA), and global urban expansion (GUE) was carried out. The characteristics of urban expansion for five representative cities including Kabul, Lhasa, Lijiang, Thimphu, and Xining were remarkably different. Based on the results of incremental analysis and the spatial difference of the ISA, it was found that the GAIA dataset at a 30-m spatial resolution could provide better ISA information than the others in characterizing urban expansion in the mountainous region. Subsequently, the changes in the urban area were analyzed using the GAIA dataset from 1993 to 2018. In general, human settlements had grown, with the transformation of small villages into larger towns and some towns into major cities. Urban expansion would continuously intensify the contradictions between human activity and sustainability and exert a more significant impact on the fragile ecosystem in the HKH region. More attention should be paid to the impact of urbanization on the fragile mountainous ecosystem.

**Keywords:** impervious surface area; urban form; Landsat images; sustainable development; urban planning

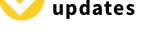



## 1. Introduction

Due to serious climate change and increasing human activity, more and more concerning issues, such as biodiversity loss, carbon emissions, water shortages, extreme weather, and environmental pollution in the mountainous Hindu Kush Himalayas (HKH) region have sparked much debate [1–8], which has the potential to provide scientific references for the sustainable development of the ecosystem [9–11]. Among human activities, the complex urbanization process with increasing urban populations results in the exploitation of natural resources, the loss of agricultural land, and the loss of biodiversity, and therefore threatens environmental sustainability [12,13]. As a concentration of buildings and infrastructure, industrial and commercial activities, and population, the expansion of urban land is often used as a critical indicator to measure the United Nation's Sustainable Development Goals (SDGs) regarding its profound effects on ecosystem services [10,14,15]. Indeed, a disorderly expansion of urban land tends to increase the ecological risk in an ecologically fragile area [16]. In the HKH region, urbanization is driven by complex local, regional, and international migration patterns. Accordingly, the exposure to a pre-existing hazard risk will increase, and ecosystem services will be weakened as urban land expands in areas susceptible to multiple hazards such as landslides and debris flows [9]. The knowledge about the temporal and spatial characteristics of urban expansion is helpful in coordinating the relationship between urbanization and the environment in such an ecologically fragile area, which makes it urgent to carry out a detailed understanding of the urban land expansion over space and time [13,17].

Accurate information about the dynamics of urban land is very necessary in order to understand the ongoing urbanization and to quantify its pace and magnitude in the future for sustainable development purposes [11,18,19]. However, surveys are difficult to carry out, and the feasible mapping of urban land is still challenging, resulting in that relevant data on the dynamics of urban areas have been deficient over the past decades [20], especially in the HKH region. Artificial structures with impermeable characteristics such as building roofs, roads, and paved areas can be recognized as impervious surface areas (ISA), which can be extracted in time using remote sensing technology to represent urban land [21]. Currently, there have been several global ISA datasets available to analyze spatio-temporal characteristics and the rule of the changes in urban land for urban management and planning [22]. Due to their distinct methods of sample collection, classification, post-processing, and accuracy assessment, there are differences between these products followed by large discrepancies in the global/regional ISA estimation [23]. Despite that, they can provide data support to quantitatively evaluate urban expansion in the HKH region to some degree.

From the perspective of sustainable development, it is meaningful and urgent to analyze the dynamics of urban expansion in the HKH region. Here, we evaluated the data quality of several ISA datasets from the characteristics of the form of urban expansion and the difference of the changes in the ISA areas of five typical cities in the HKH region. Furthermore, to understand the role of urbanization in the ecologically fragile region, the spatio-temporal pattern of urban expansion was also quantitatively described with the high-quality ISA dataset.

## 2. Study Area and Data

### 2.1. Study Area

The HKH region extends 3500 km over all or parts of eight countries, including China, Bangladesh, Bhutan, India, Myanmar, Nepal, Afghanistan, and Pakistan, and covers an area of more than 4.3 million square kilometers (Figure 1). With the elevation ranging from less than 500 m to more than 8000 m and characterized by four global biodiversity hotspots, the region has been recognized as one of the most dynamic, fragile, and complex mountain systems in the world. The source of ten of the major rivers in the world, such as the Brahmaputr, the Ganges, the Mekong, the Yangtze and the Yellow River are from the HKH region's agglomerations of snow and ice, and more than 1.3 billion people live in these downstream river basins.

To evaluate the quality of the ISA datasets, five representative cities were selected, including Kabul, Lhasa, Lijiang, Thimphu, and Xining (Figure 1). (1) Kabul, the capital of Afghanistan, is located high up in a narrow valley between the Hindu Kush mountains and bounded by the Kabul River. (2) Lhasa, the administrative capital of the Tibet Autonomous Region of China, has an average elevation of 3656 m and is surrounded by towering mountains. (3) Lijiang is within the region encompassed by the Hengduan Mountains where the Qinghai-Tibet Plateau and Yunnan-Guizhou Plateau converge, and has substantially more mountains over 2000 m than plains. (4) Thimphu is in the western central part of Bhutan and ranking the fifth highest capital in the world in terms of elevation. Thimphu was selected as a representative city for it is the capital and the largest city in Bhutan. (5) As the capital of Qinghai province in China, Xining is the largest city in terms of population and urban area on the Qinghai-Tibet Plateau, which lies in the Huangshui River valley and has a cool climate on the borderline between cool semi-arid and humid continental climates.

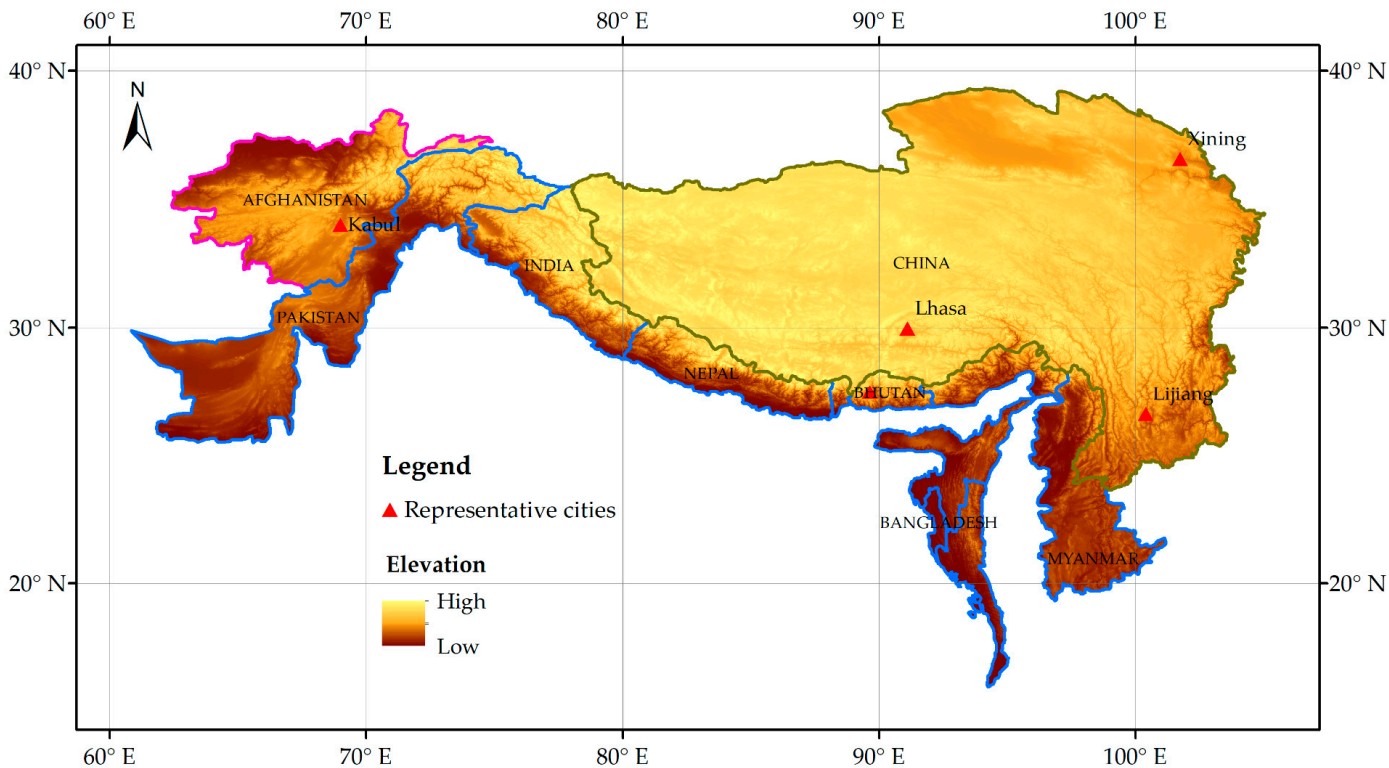

**Figure 1.** Map of the location of the HKH region and the selected cities used to evaluate the artificial impervious area products. (The elevation derived from GTOPO30, a global digital elevation model (DEM) with a horizontal grid spacing of 30 arc seconds from https://e4ftl01.cr.usgs.gov/MEASURES/SRTMGL1.003/2000.02.11/index.html (accessed on 15 December 2021). The region borderline with ginger pink is Afghanistan, the borderline with blue indicates the region including Pakistan, India, Nepal, India, Bhutan, India, Bangladesh, and Myanmar from the northwest to the southeast, and the region with dark olivenite refers to part of China).

### 2.2. Data Sources

From a practical standpoint, a total of four freely available global fine-resolution ISA products with long time series and high spatial resolution were adopted in this study (Table 1).

**Table 1.** Overview of the ISA datasets selected in the research.

| Name of Dataset | Spatial Resolution | Time of Period | Abbreviation | Definition of Urban Land | Method | Data Source | Reference |
|---|---|---|---|---|---|---|---|
| Global artificial impervious areas | 30 m | 1985–2018 | GAIA | Artificial impervious surfaces | Conventional maximum likelihood classifier, the J4.8 decision tree classifier, the random forests ensemble classifier and the support vector machine | Landsat images | [24] |
| Global annual urban dynamics | 30 m | 1985–2016 | GAUD | Impervious surface | Spectral index-based method | Landsat images, DMSP OLS NTL | [19] |
| Global impervious surface area | 30 m | 1972–2019 | GISA | Impervious surface | Machine learning | Landsat images | [23] |
| Global urban expansion data | 1 km | 1992, 1996, 2000, 2006, 2010, 2016 | GUE | Urban land | A fully convolutional network | MODIS, Landsat images, DMSP OLS * | [4] |

\* DMSP OLS: Defense Meteorological Satellite Program's Operational Linescan System; MODIS: Moderate Resolution Imaging Spectroradiometer; NTL: nighttime light.

(1) The global annual impervious area (GAIA) was downloaded from http://data. ess.tsinghua.edu.cn (accessed on 8 March 2022) and it could provide the annual dynamics of global urban details at a 30-m spatial resolution from 1985 to 2018 [24]. (2) The global annual urban dynamics (GAUD) at a 30-m spatial resolution was freely available from https://figshare.com/articles/dataset/High_spatiotemporal_resolution_mapping_ of_global_urban_change_from_1985_to_2015/11513178/1 (accessed on 10 March 2022) [19]. (3) The global impervious surface area (GISA) dataset was downloaded from http://irsip. whu.edu.cn/resources/dataweb.php (accessed on 9 March 2022) [23]. (4) The global urban expansion (GUE) was downloaded from the website https://doi.pangaea.de/10.1594/ PANGAEA.892684 (accessed on 11 March 2022). The remote sensing data sources for the GUE data were sourced from DMSP-OLS nighttime light data, the MODIS NDVI (normalized difference vegetation index), the NOAA-AVHRR NDVI, as well as the LST (land surface temperature) [4].

### 2.3. Methods

The landscape metrics i.e., the shape and distribution of urban patches, are meaningful for land-use products by characterizing and quantitatively describing the urban form, and visually comparing sample pixels with Google Earth is very practical in measuring the accuracy of urban land-use classification [15,25,26].

To evaluate the quality of the ISA datasets in the HKH region, landscape metrics such as impervious surface morphology of the four ISA datasets was visually compared to sample pixels with Google Earth in detail. Firstly, five representative cities were selected, and corresponding samples were collected from a Google Earth historical high spatial resolution image to make a visual and detailed evaluation of the cities' urban form. Following the analysis of the spatial difference of urban expansion, an incremental analysis of the ISA datasets was comparatively undertaken. Moreover, some existing research results with regard to land use and land cover change were used to measure the comparison of the four ISA datasets. In the end, the spatiotemporal analysis of urban expansion in the HKH region was quantitatively carried out with the high-quality dataset.

## 3. Results and Discussions

### 3.1. Reliability Analysis of the ISA Datasets

3.1.1. Spatial Differences of Urban Expansion Represented by the ISA Datasets

To evaluate the quality of the ISA datasets, the impervious surface morphology of the four ISA datasets featured by urban spatial expansion was first analyzed in detail. Figure 2 shows the characteristics of urban expansion of the five representative cities between 2000 and 2016. For Kabul, a city surrounded by low but steep mountain ranges, the characteristics of urban spatial expansion described by GAIA were more detailed than those of the other datasets. More ISA expansion between 2000 and 2016 was found in the north quarter of Kabul City than the south quarter, in accordance with the actual situation that the south bank of the Kabul River through the city is the old quarter, and the north bank is the new quarter. In comparison, the GAUD dataset showed more ISA in 2016, while the GISA showed more ISA in 2000. GUE only depicted the city center. In fact, postwar refugees, the return of internally displaced inhabitants to the city, and rural migration have contributed to a large proportion of the total population, leading to the growth of informal settlements with a high demand for the construction of plots in Kabul [27]. With the opening of well-developed highways and railways as well as the issue of relevant national policies, the model of urban spatial structure of Lhasa has gradually changed from a compact one to a leapfrogged urban expansion one [28]. Generally, all of the ISA datasets showed that the urbanized area of Lhasa was roughly concentrated in the urban areas, showing a trend of monocentric urban structure from 2000 to 2016. The characteristic of urban areas by GAIA and GAUD agreed more with the imagery than did the other datasets. However, most ISA of GAUD was built after 2000, with no efficient extraction of the ISA in 2000. Interestingly, the basic urban shape wasn't reflected by GISA at a 30-m spatial

resolution. GUE at a 1-km spatial resolution was too coarse to describe urban expansion. For Lijiang City, which has the higher terrain in the northwest than in the southeast, the four datasets could represent the urban shape. The products at 30-m spatial resolution gave more details. With steep elevations, Thimphu is limited to the east and west and runs roughly halfway up the surrounding hillsides, making it more difficult for ISA extraction using remote sensing technology. Surrounded by mountains, the urban area of Xining is limited in areal extent and restricted on a narrow strip of land, with the length much greater in longitude than in latitude. The urban shape was well expressed by the ISAs from GAIA, GAUD and GISA. The GUE dataset only showed the main body of the city. Thus, the land expansion in Xining showed a strip form and had small patches restrained by the narrow plains, which was consistent with the results reported by Deng et al. [29] that the population of Xining was centralized and monocentric due to topography. Concretely, compared to the other datasets, GAIA and GISA had better performances in urban areas and showed good impervious surface patterns.

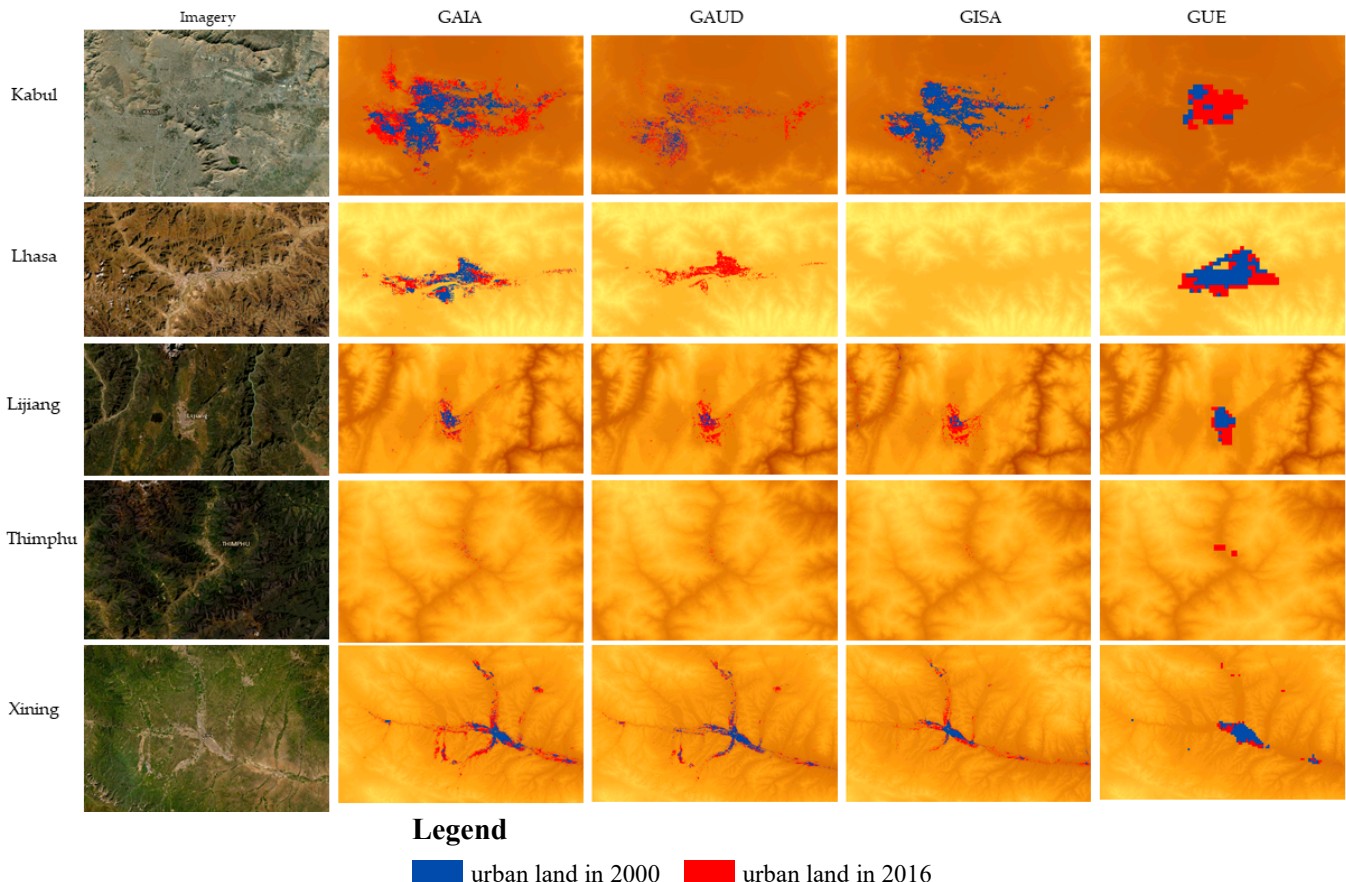

**Legend**

■ urban land in 2000     ■ urban land in 2016

**Figure 2.** Urban expansion characteristics of the five representative cities between 2000 and 2016 shown by four ISA datasets.

Although Landsat images were used as the main data source to produce the four ISA datasets, different algorithms and ancillary datasets were adopted in this study (Table 1). In addition, both the missing of Landsat archives (especially due to clouds) and sparse training samples of impervious surfaces in such mountainous regions lowered the applicability and reliability of the ISA products with a long time series. Considering the differences, concerns about the confusions in spectral analysis should include the relationship between the bright impervious surface and the desert soil and the relationship between the dark impervious surface and the dry farmland. Fortunately, the combined use of optical and Synthetic Aperture Radar (SAR) images would be effective in improving the land-cover classification and impervious surface estimation by clarifying confusions between the bright impervious

surface and bare soil, between the dark impervious surface and bare soil, and between the shaded area and the water surface [30,31]. In cities surrounded by barren mountains, such as Kabul and Lhasa, the extracted ISAs by GAIA had a better performance compared to those of by the other datasets, which could be attributed to the fact that only annual GAIA was mapped using the nighttime light data and the Sentinel-1 SAR data as ancillary datasets [24].

To some extent, the optimal ISA to represent the urban expansion in the HKH region was from the GAIA dataset.

### 3.1.2. Incremental Analysis of the ISA Datasets

Although the HKH region had experienced dramatic urban expansion, the urban land area since 1992 interpreted by the four ISA datasets displayed obvious differences due to different methods and data sources of remote sensing (Figure 3). During the period, the extent of ISA interpreted by GAIA, GAUD, GISA, and GUE was increased by 1.61, 2.80, 2.37, and 6.19 times, respectively. Even at the same spatial resolution of 30 m, the ISAs of GAUD were 801 km$^2$, approximately half of GAIA and GISA in 1992. The ISAs derived from GUE at a spatial resolution of 1 km were just 441 km$^2$. Eight years later (in 2000), the increased ISA area interpreted by these datasets was 448, 853, 581, and 1591 km$^2$, respectively. Furthermore, the total ISAs of the GAIA were close to those of the GISA and GUE. From 2000 to 2010, the increase in ISAs interpreted by GAIA, GAUD, GISA, and GUE was about 664, 816, 1541, and 922 km$^2$, respectively. On the contrary, the increase in ISAs was 1406, 570, 1435, and 216 km$^2$ from 2010 to 2016.

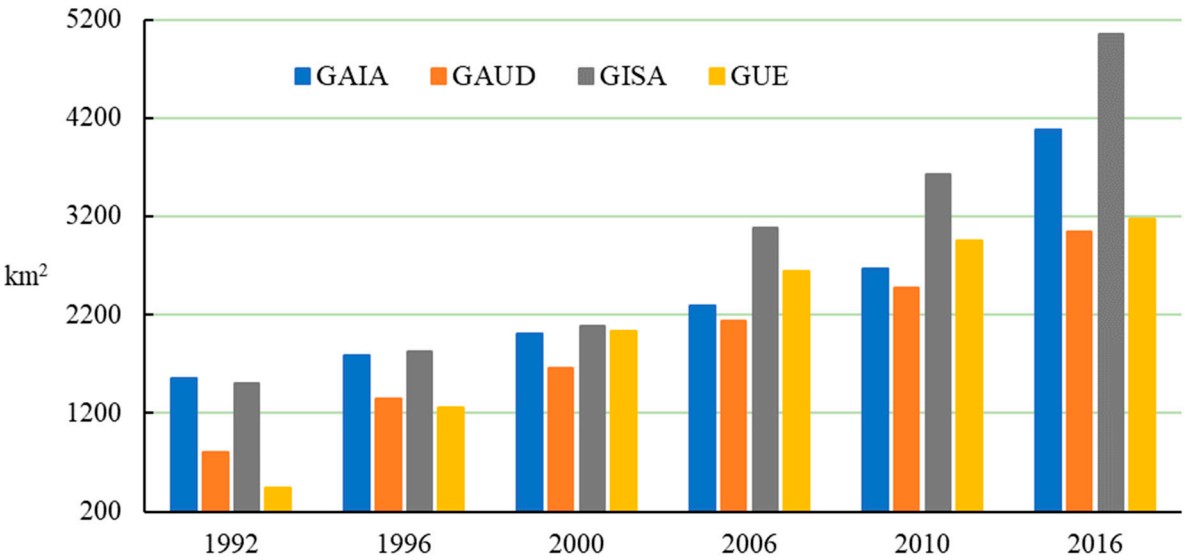

**Figure 3.** ISA changes in the HKH region interpreted by the four ISA datasets in representative years 1992, 1996, 2000, 2006, 2010, and 2016.

In terms of the representative cities, the discrepancy of the increased urban-land area represented by the four ISA datasets was also obvious (Figure 4). The land areas of Kabul City, Lijiang City, and Thimphu City in 2000 interpreted by GAIA and GISA were very close. However, the subsequent increase in ISA was remarkably different. The amount of ISA expansion of Kabul and Xining interpreted by GAIA in 2016 was more than that of the other datasets. In 2000, 2010 and 2016, the ISAs derived from the GAIA dataset were 118.3 km$^2$, 158.4 km$^2$, and 268.4 km$^2$ for Xining City, respectively, and 123.5 km$^2$, 155.3 km$^2$, and 233.8 km$^2$ for Kabul City, respectively, which is very different from the existing research results. For instance, in 2000, 2010, and 2017, the artificial surface area of Xining City was 93.81 km$^2$, 113.34 km$^2$ and 128.05 km$^2$, respectively; and the artificial surface area of Kabul was 174.73, 212.66 and 208.98 km$^2$, respectively [32]. Comparatively, the urban land of Xining City increased from 86.7 km$^2$ to 231.0 km$^2$ from 2000 to 2017 [11]. One underlying

reason was that the spatial resolution of GAIA was 30 m, higher than the resolution of data used in the existing studies, showing more details about tiny buildings. The ISA of Lhasa City in 2016 from GAIA was 98.92 km$^2$, which was close to the previous conclusion that the built-up area in Lhasa City reached 70.29 km$^2$ in November 2015 [28]. The ISA from the other datasets was far less than 70.29 km$^2$, and did not effectively reflect the characteristics of the Lhasa urban layout. Interestingly, there was no value derived from GISA for the urban land of Lhasa City. The ISAs extracted from the datasets for Thimphu City were very small, and the urban expansion was also not obvious, since it runs roughly halfway up the surrounding hillsides.

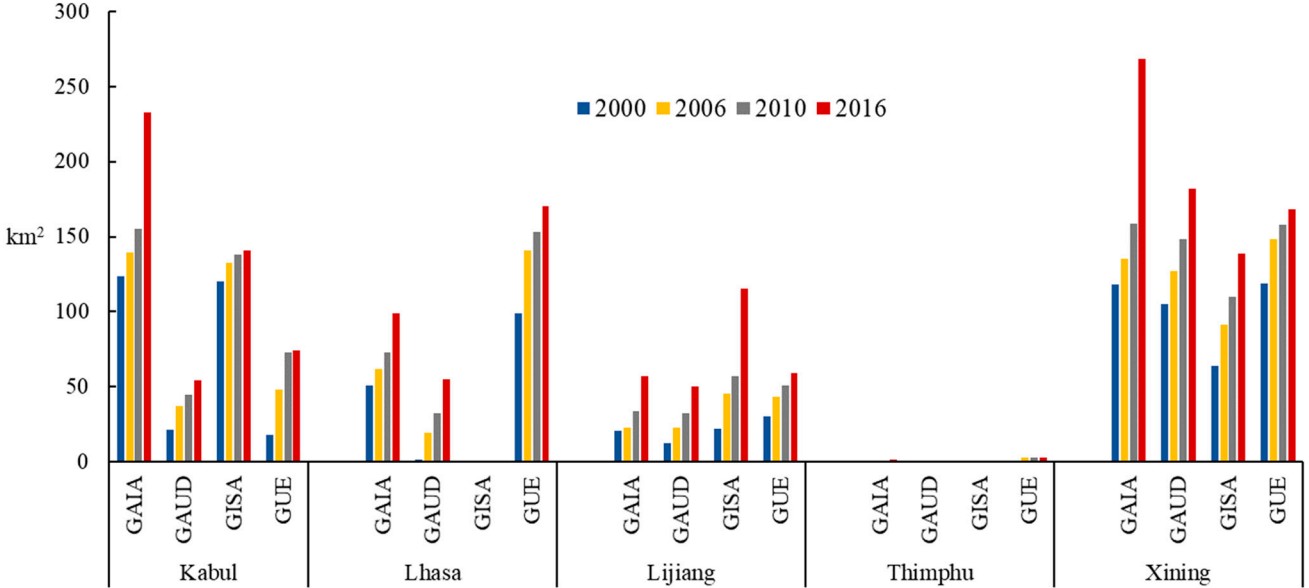

**Figure 4.** Changes in ISA area of the five representative cities from 2000 to 2016 extracted by the four ISA datasets.

Generally, high overall accuracy (OA) was achieved for these datasets in global validation, with 76% (1985–2000) and 82% (2000–2015) for GAUD in humid regions, higher than 90% for GAIA, the F-score of 0.954 for GISA, and 90.9% for GUE. At the globe scale, the ISAs derived from GISA were slightly larger than that from GAIA and GAUD, which may be attributed to the fact that the ISA extent of GAIA and GAUD were constrained by a predefined urban mask, but not GISA [23]. Indeed, the ISAs from GISA were larger than those from GAIA and GAUD in the HKH region. These findings suggest that the ISA interpreted by GUE could not describe the mountainous urbanization in the HKH region. Due to the mountain setting with steep terrain, limited resources, and frequent natural hazards, the process of urbanization is restrained, and most of the HKH region is still the least developed [33]. Based on the above, it was reasonable that the ISA extent was 1.61 times that of 1992 in 2016, that is, the GAIA dataset might be better for such a mountainous region.

Following the analysis of the spatial difference of urban expansion and the increased ISA in the representative cities and the HKH region, the GAIA dataset was found to be more suitable to understand the urbanization in such a mountainous region compared to the other datasets.

### 3.2. Analysis of Urban Changes in the HKH Region

Due to different elevation and different state systems, it was unreasonable to analyze the urban changes integrally. Based on the elevations and different national territories, the ISA dynamics of three divided areas and the entire region had been analyzed with the GAIA dataset to learn about urban expansion. The three divided areas referred to the following:

(1) the country in turmoil, namely Afghanistan; (2) the countries of Pakistan, India, Nepal, India, Bhutan, India, Bangladesh, and Myanmar, from the northwest to the southeast along Hindu-Kush; and (3) the regions belonging to China as part of China (Figure 1). The ISAs ranged from 1614.6 km$^2$ in 1993 to 4314.7 km$^2$ in 2018 for the entire HKH region, with an average growth rate of 0.07 (Figure 5 and Table 2). Regarding the divided areas, the average annual growth rates of ISA from 1993 to 2018 were 0.05, 0.05 and 0.09 for Afghanistan, PAK-BAN-BTN-IND-MMR-NEL and part of China, respectively, suggesting that urban expansion in Part of China was faster compared to the other areas (Table 2 and Figure 6). Despite the fact that political chaos and security turbulence prevailed in Afghanistan during the period, out-migrants from the mountainous countryside and Afghan refugees back from neighbouring countries resulted in rapid unplanned ISA expansion, especially on the northern plains. In Part of China, cities and towns with relatively low terrain and wide spaces suitable for human activities had attracted an increasing number of immigrants. Furthermore, industrialization and transportation also made the ISA expansion possible. For example, the railway and national roads, such as the Tibetan Railway and the Beijing-Tibet Expressway, had made Lhasa City and Xining City important transportation hubs on the Qinghai-Tibet plateau, leading to the ISA expansion of two cities with more immigrants and human activities along the transportation lines. In general, ISA expansion occurred primarily in the northeast, southeast and southern part of the Qinghai-Tibet plateau; comparatively, there was no ISA in the central and western regions, which was consistent with the conclusion about the level of urbanization on the Qinghai-Tibet plateau [34]. The southeast of Part of China is mountainous with abundant biological resources and is experiencing slow but conspicuous expansion of ISA. Moreover, national policies, i.e., the development of western regions, the new-type planning of urbanization, as well as the Belt and Road initiative, could also stimulate the economic activities and the level of urbanization on the Qinghai-Tibet plateau [34]. In Part of China, the expansion of construction land was at the expense of farmland and grassland, and urbanization made the interactions between different land uses more significant. Consequently, the inflow of these people would negatively influence the environment, safety, transportation and cultural heritage [28]. For PAK-BAN-BTN-IND-MMR-NEL, the expansion of ISA lied in the fact that the migration from rural to urban areas in the lower hills of the Himalayas had been proceeding at a rapid rate, resulting in an unplanned, seasonal makeshift and informal urbanization in a growing number of small towns around religious centers [35,36]. Regardless of their size, small urban centers in the HKH region had relevant functions for the surrounding rural areas and served as hotspots for economic development, often accompanied by a decline in the local agriculture and a degradation of the environment [37]. In addition, urban and rural expansion combined with increased road infrastructure had driven the agricultural land into the forest [35]. As ISA expanded, more migrants tended to live in those places with improved public infrastructure, which leads to the expansion of settlements into agricultural land and forest. Thus, a conclusion could be drawn that the area of human settlements in the HKH region had expanded, with the development of some small villages into larger towns and some former towns into major cities. As the topography and climate suitable for human settlement are limited in the HKH region, the distribution of the population is tightly restricted, and the concentrated areas for current migration and urban expansion are highly vulnerable areas susceptible to multiple hazards [9], suggesting that the process of urbanization will keep putting more people on the line. In addition, the growth of the population, the migration from rural settlements, and the limited space suitable for construction and urbanization would exert more serious impacts on the fragile ecosystem and create a set of vulnerabilities, such as water shortages, and poor-quality water, sanitation and drainage, and human activities in landslide or flood-prone areas [8,28,34,38–43], which would intensify the contradictions between urbanization and sustainability in the HKH region.

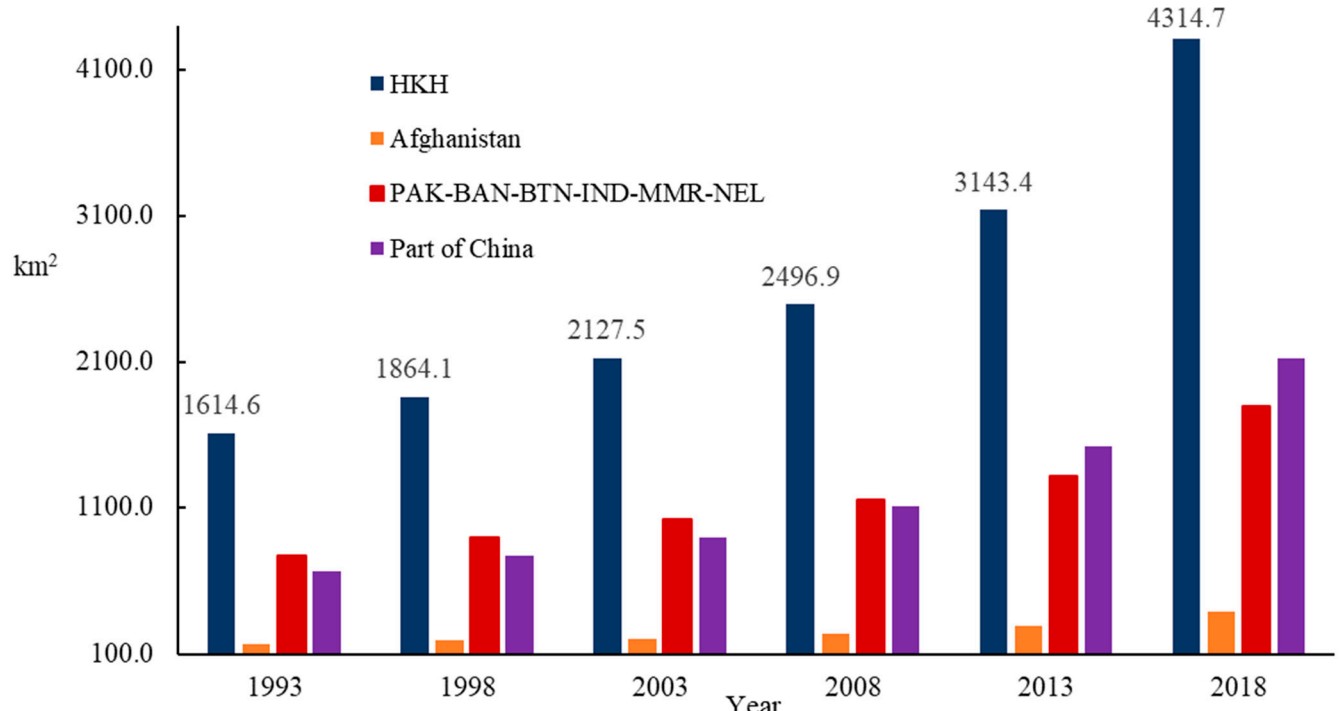

**Figure 5.** Areas of ISA for the entire region and three divided areas from 1993 to 2018.

**Table 2.** Average annual growth rate of ISA from 1993 to 2018.

| Region \ Time Period | 1993–1998 | 1998–2003 | 2003–2008 | 2008–2013 | 2013–2018 | 1993–2018 |
|---|---|---|---|---|---|---|
| HKH | 0.03 | 0.03 | 0.03 | 0.05 | 0.07 | 0.07 |
| Afghanistan | 0.03 | 0.02 | 0.03 | 0.05 | 0.07 | 0.05 |
| PAK-BAN-BTN-IND-MMR-NEL | 0.03 | 0.03 | 0.03 | 0.03 | 0.07 | 0.05 |
| Part of China | 0.03 | 0.03 | 0.05 | 0.08 | 0.08 | 0.09 |

With the advances of urbanization and economic development, there is still a high rate of poverty and marginalization attributed to socioeconomic inequalities, separate policies, and environmental challenges in the HKH region. The issues of concern included how to improve the quality of urban development and how to strengthen the protection of the environment while carrying out healthy urbanization [16,44]. A pivotal measure is to promote scientific urban planning with proper urban-land development in such a mountainous region for sustainable development [21,45,46]. For example, Nepal government's policy of 'developing smart cities in the hills' stimulated urbanization and improved human life [13]. For a city whose pattern of spatial development mainly depends on topography and transportation, knowledge about the effects of urbanization is very useful for urban planning and developing reasonable strategies [46,47]. However, at all events, the detailed and in-depth assessment of multi-hazard susceptibility and exposure are prerequisites for hazard risk mitigation for small settlements spread over the region [48]. The complexity characterized by significant regional complexity, the heterogenous landscape, and cultural and social diversity along with the extreme and fragile natural conditions in the mountainous region might get in the way of developing regional collaborative policies or conservation interventions [35]. If urban development can take advantage of geographical location, unique tourist sites, and historical and cultural components, harmonious urban brands with mountainous areas will be established [49]. In general, decisive policy responses such as planning, regulatory and pricing tools are requisites for avoiding further

costly lock-in along with the urbanization development in this region [26,50]. Only on the basis of a systematic assessment of the urban expansion on the ecosystem should effective measures such as optimization of the urban layout, the exploration of high-quality urban development patterns and the promotion of regional coordinated development be taken to maintain the sustainable development of society and ecology and to achieve the harmony between human activities and ecological protection in the HKH region.

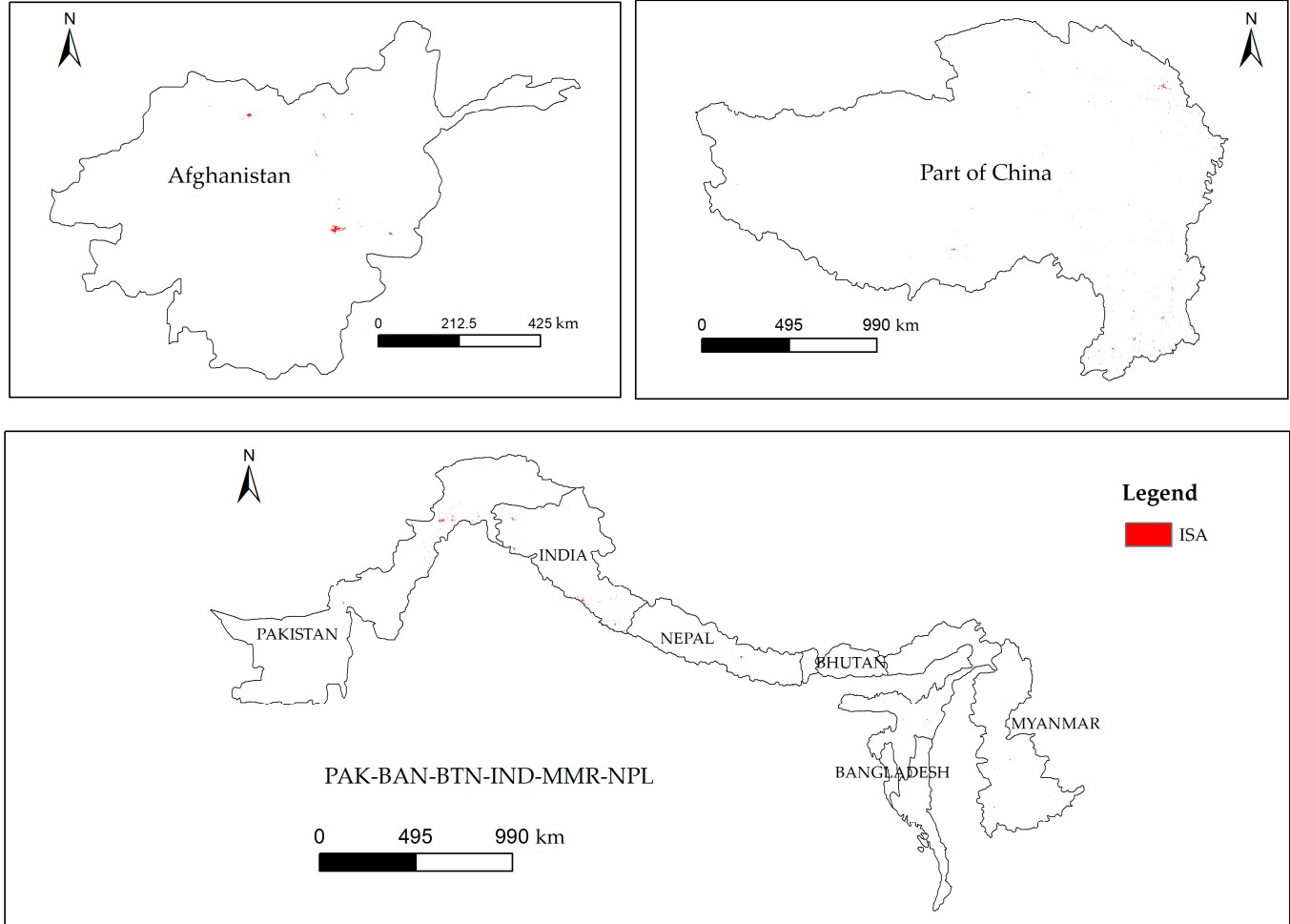

**Figure 6.** Diagram of the spatial distribution of the ISA for the three divided areas.

## 4. Conclusions

As the nexus of human activities and economic activities, urbanization has profound environmental impacts which can pose great challenges to the fragile mountain ecosystem in the HKH region. According to the reliability evaluation of four global fine-resolution ISA products (GAIA, GAUD, GISA and GUE), the GAIA dataset that used the nighttime light data and the Sentinel-1 SAR data as ancillary datasets could provide better information on ISA to characterize urban expansion in the mountainous regions from the perspective of the incremental analysis of the ISA and the spatial characteristics of the urban expansion. Based on the analysis of the urban changes using the GAIA dataset, the total urban area increased by 2700.1 km$^2$ from 1993 to 2018. As ISA expanded, the increasing livelihood opportunities and convenient public infrastructure became more attractive for young adults from rural settlements or from the hills to the lower plains. Urban land had grown, along with the transformation of many small villages into larger towns and some former towns into major cities. Due to the topography and climate, the concentrated areas for current migration and urban expansion are highly vulnerable areas susceptible to multiple hazards in the HKH region, suggesting that the process of urbanization would keep putting more people

on the line. Furthermore, urbanization would exert more serious impacts on the fragile ecosystem and face limited resources such as water shortages. The contradictions between urbanization and ecological sustainability would be intensified and have more negative impacts on the fragile ecosystem in the HKH region. A brief analysis of the spatial-temporal characteristics of urban land change was undertaken and more existing achievements were referred to support the discussion about the effects on the fragile ecosystem. To maintain sustainable development in such a mountainous region, more work should be carried out to make more scientific urban planning such as formulating new urbanization strategies and building smart cities in the future.

**Author Contributions:** Z.C. & C.F.: Investigation, Conceptualization, Formal analysis, Writing—original draft, Methodology, Supervision. Z.Z. & M.Z.: Investigation, Visualization. Z.S.: Conceptualization, Methodology, Formal analysis. All authors have read and agreed to the published version of the manuscript.

**Funding:** This research was funded by the National Natural Science Foundation of China (No. 31961143012), the Horizontal Scientific Research Project (No. 21ZH285), and the Innovation Training Program for College Students in Nantong University (No. 2022193).

**Data Availability Statement:** Publicly available datasets were analyzed in this study. This data can be found here: http://data.ess.tsinghua.edu.cn, https://figshare.com/articles/dataset/High_spatiotemporal_resolution_mapping_of_global_urban_change_from_1985_to_2015/11513178/1, http://irsip.whu.edu.cn/resources/dataweb.php, and https://doi.pangaea.de/10.1594/PANGAEA.892684.

**Acknowledgments:** We would like to deeply appreciate the help from Bin Lin (Fujian Normal University, China). We also benefit from the anonymous reviewers' constructive comments and suggestions.

**Conflicts of Interest:** The authors declare that they have no known competing financial interests or personal relationships that could have appeared to influence the work reported in this paper.

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
