# Peer review of "Spatiotemporal Analysis of Urban Expansion in the Mountainous Hindu Kush Himalayas Region"

_land, doi:10.3390/land12030576_

Round 1

Reviewer 1 Report

Overall, reliability evaluation of four global fine-resolution impervious surface area products, global annual impervious area, global annual urban dynamics, global impervious surface area, and global urban expansion, was carried out. I believe that the manuscript could benefit from additional attention to presentation (in the text), and some additional methodological details and justifications, to strengthen the conclusions and improve clarity. The following issues need to be addressed.

- A clear theoretical framework should be summarised after the literature review. On the urban compactness to ecosystem services in a rapidly urbanising metropolitan area, this article can be referenced.

- This paper is much descriptive. The theoretical thinking in the paper should be strengthened. Which theoretical or academic arguments are the authors aiming to examine or discuss? What new theories or theoretical knowledge this paper can add in the existing literature by looking the local cases?

- Dialogue and comparison with other studies cannot be missing.

- What new global knowledge can this paper contribute to the existing international literature? It is recommended that the authors conclude with a discussion of the applicability of the method to other areas.

- Takeaway for Practice is also encouraged to be included in this paper. It should be clear enough to present your policy recommendations for both local and international practice.

- Limitations are not well summarised.

Author Response

General comments

Overall, reliability evaluation of four global fine-resolution impervious surface area products, global annual impervious area, global annual urban dynamics, global impervious surface area, and global urban expansion, was carried out. I believe that the manuscript could benefit from additional attention to presentation (in the text), and some additional methodological details and justifications, to strengthen the conclusions and improve clarity. The following issues need to be addressed.

Response: Thank you very much for your kind comments.

In the revised version, all your suggestions were fully considered and the revisions were made respectively. You can find the revised part marked in red in the revised manuscript.  

Point 1: A clear theoretical framework should be summarised after the literature review. On the urban compactness to ecosystem services in a rapidly urbanising metropolitan area, this article can be referenced.

Response 1: Thank you very much for your kind comments.

The work about drivers and impacts on the ecosystem are referenced and a clear theoretical framework was summarised in the manuscript. and more existing achievements were referred to support our work. Here, what we’re focused on was the reliability evaluation of four global fine-resolution impervious surface area products in the HKH region. And the spatial-temporal characteristics of the urban land change were also talked about.

Point 2: This paper is much descriptive. The theoretical thinking in the paper should be strengthened. Which theoretical or academic arguments are the authors aiming to examine or discuss? What new theories or theoretical knowledge this paper can add in the existing literature by looking the local cases?

Response 2: Thank you very much for your kind comments.

In the ecologically fragile Hindu Kush Himalayas (HKH) region, serious climate change and increasing human activities often lead to biodiversity loss, carbon emissions, water shortages, extreme weather, and environmental pollution. Particularly, the role of human activities is noteworthy. While the effect about the urban land change on the ecosystem is rarely talked about as few reliable data can be used to represent the urban sprawl. In the manuscript, we carried out the reliability evaluation. To a degree, the detailed analysis about the reliability of the four datasets would give dataset producers more information about methods and auxiliary data utilization in such mountainous region. And with the reliable dataset, some research about urban change on ecosystem can be carried out in depth.

Point 3: Dialogue and comparison with other studies cannot be missing.

Response 3: Thank you very much for your kind comments.

In the manuscript, more existing achievements were referred to support our work or provoke more discussion. We hope more detailed work about the ecosystem assessment impacted by urban expansion would be carried out as reliable urban land data can be derived from remote sensing data. 

Point 4: What new global knowledge can this paper contribute to the existing international literature? It is recommended that the authors conclude with a discussion of the applicability of the method to other areas.

Response 4: Thank you very much for your kind comments.

As more and more attentions are paid to the mountainous HKH region, reliable data are very valuable for further study about the effect of human activities and climate change on ecosystem. In our study, we just talked about the urban land data and it would provide more knowledge about the urban sprawl.

Point 5: Takeaway for Practice is also encouraged to be included in this paper. It should be clear enough to present your policy recommendations for both local and international practice.

Response 5: Thank you very much for your kind comments.

In the manuscript, more existing achievements were referred to recommend relevant policy. Frankly, the manuscript is very detailed in terms of data evaluation and spatial-temporal characteristics of the urban land change in the HKH region. In the future, work would be carried out to talk about policy recommendations for both local and international practice.

Point 6: Limitations are not well summarised.

Response 6: Thank you very much for your kind comments.

To some extent, the manuscript adopted the method popular in the relative researches to evaluate the four datasets. It should be thin if we just carried out the quality assessment of these ISA datasets. So, the analysis of spatial-temporal characteristics was done preliminarily due to limited paper length.  The main limitation is that the manuscript lack depth though we talked about the reliability and spatial-temporal characteristics of urban sprawl. So the limitations are summarised briefly in the manuscript.

Reviewer 2 Report

General Comments

Urban sprawl is hot issue in the world. The novel point of the manuscript is to find the suitable ISA dataset by comparing the different ISAs datasets in the mountainous Hindu Kush Himalayas region. The findings are valuable to enhance the precision of mountainous urban sprawl study. However, how urban sprawl is too simple design, and process, driving factors and impacts on fragile ecosystem mentioned in manuscript lack data validation. It is necessary to construct an indicator to analyze how urban sprawl impacts on fragile ecosystem in a separate section to make the conclusions convincing.

In addition, it is recommended to keep the same terms (urban sprawl or urban expansion) consistent back and forth in manuscript. Language needs to be further polished

Particular Comments

1. Introduction

This section needs to be improved. Authors just emphasize the data quality of several ISA datasets, but drivers and impacts on the ecosystem related issues are not addressed which appear in the conclusions.

2. Study area and Data

Labels of elevation are reversed in Figure 1.

Data source of GAIA in Table 1 cannot match with description in line 197-198.

It is unclear that how four ISA datasets were visually compared to sample pixels with Google Earth in detail in line 127-128.

3. Results and discussions

Discussion is essential part for making your conclusions convinced, so I suggest that the discussion be listed separately. There few data to verify your results, so I suggest use data in study area to make your results convincing.

There is a confused description in line 224-225, which can be shown in Figure 3 instead of Figure 4.

Please give the basis of dividing three areas in line 270-274.

Compass markers in Figure 6 is missing in line 353.

4. Conclusions and policy implications

Most conclusions cannot make sense in lines 356-357, 364-369. I cannot find any information proved in the manuscript in lines 356-357. I also suggest add population data and classification of cities to verify your conclusion in lines 364-365, 365-366. According to the results, it is necessary to propose targeted strategies to promote sustainable urban expansion in mountainous region.

I suggest the authors rewrite this section and results section to make the conclusions convincing.

Author Response

General Comments

Urban sprawl is hot issue in the world. The novel point of the manuscript is to find the suitable ISA dataset by comparing the different ISAs datasets in the mountainous Hindu Kush Himalayas region. The findings are valuable to enhance the precision of mountainous urban sprawl study. However, how urban sprawl is too simple design, and process, driving factors and impacts on fragile ecosystem mentioned in manuscript lack data validation. It is necessary to construct an indicator to analyze how urban sprawl impacts on fragile ecosystem in a separate section to make the conclusions convincing.

In addition, it is recommended to keep the same terms (urban sprawl or urban expansion) consistent back and forth in manuscript. Language needs to be further polished

Response: Thank you very much for your kind comments.

In the revised version, the drivers and impacts on the ecosystem related issues were present in the Introduction section. All your suggestions were fully considered and the revisions were made respectively. The terms, urban sprawl or urban expansion, were kept consistent back and forth in the manuscript. You can find the revised part marked in red in the revised manuscript.

In the manuscript, more existing achievements were referred to support the impacts of urban expansion on fragile ecosystem. We hope more detailed work about the ecosystem assessment impacted by urban expansion will be carried out in the future as reliable urban land data can be derived from remote sensing data. 

Particular Comments

Point 1:  1. Introduction

This section needs to be improved. Authors just emphasize the data quality of several ISA datasets, but drivers and impacts on the ecosystem related issues are not addressed which appear in the conclusions.

Response 1: Thank you very much for your kind comments.

We have addressed the drivers and impacts on the ecosystem related issues in the Introduction section. With the help of existing publications, the effects of urban expansion on the fragile ecosystem were widely talked about both in the Introduction, Results and Discussions, and Conclusions.

Point 2:  2. Study area and Data

Labels of elevation are reversed in Figure 1.

Data source of GAIA in Table 1 cannot match with description in line 197-198.

It is unclear that how four ISA datasets were visually compared to sample pixels with Google Earth in detail in line 127-128.

Response 2: Thank you very much for your kind comments.

Labels of elevation are right in Figure 1. Generally, the elevations of the Qinghai-Tibet plateau are higher than those in Afghanistan, Pakistan, and so on. The higher elevations with light colour and the lower elevations with deep colour make the Figure 1 look more comfortable. Data source of GAIA in Table 1 just list the main data source for datasets. Sentinel-1 SAR data helps clarifying confusions between bright impervious surface and bare soil, between dark impervious surface and bare soil, and between shaded area. Nighttime light and Sentinel-1 Synthetic Aperture Radar were used as more constraints to the algorithm to improve the initial algorithm in arid regions in GAIA datasets. As ancillary datasets, the Nighttime light and Sentinel-1 SAR datasets were not list in Table 1.

We have improved the expression to make it unclear about the comparison of four ISA datasets to sample pixels with Google Earth in detail, which is showed in red color in the revision.

Point 3:  3. Results and discussions

Discussion is essential part for making your conclusions convinced, so I suggest that the discussion be listed separately. There few data to verify your results, so I suggest use data in study area to make your results convincing.

There is a confused description in line 224-225, which can be shown in Figure 3 instead of Figure 4.

Please give the basis of dividing three areas in line 270-274.

Compass markers in Figure 6 is missing in line 353.

Response 3: Thank you very much for your kind comments.

There are few publications about the urban expansion in the HKH region. In the manuscript, we had referred to relevant research results to support the Results and Discussions. Some results were mixed with discussions.

The confusion in Line 224-225 was that the city areas in 2000 instead of in 1992, which is shown in Figure 4. We made a mistake.

Thank you for the suggestion and we give the simple basis of dividing three areas in line 270-274.

We had added the compass markers in Figure 6.

Point 4:  4. Conclusions and policy implications

Most conclusions cannot make sense in lines 356-357, 364-369. I cannot find any information proved in the manuscript in lines 356-357. I also suggest add population data and classification of cities to verify your conclusion in lines 364-365, 365-366. According to the results, it is necessary to propose targeted strategies to promote sustainable urban expansion in mountainous region.

I suggest the authors rewrite this section and results section to make the conclusions convincing.

Response 4: Thank you very much for your kind comments.

We had improved the relative expressions and made it clearer with more references.  It would be thin if we just carried out the quality assessment of these ISA datasets as it was very difficult to verify the urban expansion accurately. So, the tentative analysis of spatial-temporal characteristics of urban expansion was done due to limited paper length. In the manuscript, more existing achievements were referred to support our work. More detailed work about the ecosystem assessment impacted by urban expansion should be carried out in the future.   

Reviewer 3 Report

the innovations of this paper should be presented clearly.

Author Response

General comments

the innovations of this paper should be presented clearly.

Response: Thank you very much for your kind comments.

Your suggestions were fully considered, and the revisions were made respectively. You can find the revised part marked in red in the revised manuscript. In the manuscript, more existing achievements were referred to support our work. We hope more detailed work about the ecosystem assessment impacted by urban expansion should be carried out as reliable urban land data can be derived from remote sensing data. 

Round 2

Reviewer 2 Report

The authors have amended the most flaws in manuscript. There are still two points need improving.

Point 1: Reduce description of drivers and ecosystem related issues the authors write too much in line 45-59 in introduction section that are not key issues in manuscript.

Point 2: I still believe it is necessary to propose targeted strategies to promote sustainable urban expansion in mountainous region according to the results, and at least give governors some hints in decision making.
